# Three-dimensional covalent organic frameworks with nia nets for efficient separation of benzene/cyclohexane mixtures

Jianhong Chang[1,4], Fengqian Chen[1,4], Hui Li[1] ✉, Jinquan Suo[1], Haorui Zheng[1], Jie Zhang[1], Zitao Wang[1], Liangkui Zhu[1], Valentin Valtchev [2,3], Shilun Qiu[1] & Qianrong Fang [1] ✉

The synthesis of three-dimensional covalent organic frameworks with highly connected building blocks presents a significant challenge. In this study, we report two 3D COFs with the **nia** topology, named JUC-641 and JUC-642, by introducing planar hexagonal and triangular prism nodes. Notably, our adsorption studies and breakthrough experiments reveal that both COFs exhibit exceptional separation capabilities, surpassing previously reported 3D COFs and most porous organic polymers, with a separation factor of up to 2.02 for benzene and cyclohexane. Additionally, dispersion-corrected density functional theory analysis suggests that the good performance of these 3D COFs can be attributed to the incorporation of highly aromatic building blocks and the presence of extensive pore structures. Consequently, this research not only expands the diversity of COFs but also highlights the potential of functional COF materials as promising candidates for environmentally-friendly separation applications.

Covalent organic frameworks (COFs) represent an emerging class of crystalline porous materials that integrate reticular chemistry and dynamic covalent chemistry[1–8]. These frameworks are synthesized through reversible covalent reactions between organic modules. COF materials possess highly desirable properties, including a high surface area, exceptional porosity, and customizable pore environment, which make them highly promising for a wide range of molecular separations and other applications[9–15]. In particular, three-dimensional (3D) COFs usually feature more intricate porous structures, interconnected channels, and increased accessibility to active sites, compared to their two-dimensional (2D) counterparts. This distinguishes them by providing enhanced capabilities for molecule adsorption and separation, as opposed to 2D COFs, which rely on interlayer π-π interactions to form one-dimensional (1D) pore channels[16–20]. However, the exploration of 3D COFs has been limited due to the scarcity of organic monomers and the challenges associated with their synthesis and

structural elucidation, particularly when employing highly connected building blocks. The existing repertoire of 3D COFs mostly relies on tetrahedral (4-connected) building units derived from compounds such as tetraphenylmethane, tetraphenylsilane, and adamantane derivatives[21–27]. This constraint restricts the structural diversity and functionalization potential of 3D COFs.

Recent advancements in COF research have yielded the synthesis of COFs exhibiting unique topologies based on highly connected nodes[28–32]. One notable example is the work successfully developed a novel series of 3D covalent organic frameworks (COFs) with a unique **she** topology using $D_{3d}$-symmetric building blocks, demonstrating excellent crystallinity, porosity, and catalytic activity, thus expanding the structural diversity of COFs[33]. Another example is the work conducted by Cooper and colleagues, where they successfully incorporated organic cage molecules as 6-connected units, resulting in a 2-fold interpenetrated

[1]State Key Laboratory of Inorganic Synthesis and Preparative Chemistry, Jilin University, Changchun, People's Republic of China. [2]Qingdao Institute of Bioenergy and Bioprocess Technology, Chinese Academy of Sciences, Qingdao, Shandong, People's Republic of China. [3]Normandie Univ, ENSICAEN, UNICAEN, CNRS, Laboratoire Catalyse et Spectrochimie, Caen, France. [4]These authors contributed equally: Jianhong Chang, Fengqian Chen. ✉e-mail: postlh@jlu.edu.cn; qrfang@jlu.edu.cn

**acs** network[34]. In another significant contribution, Yaghi et al. employed cubic (8-connected) nodes to design and synthesize a series of polycubane 3D COFs with **bcu** topology[35]. Our research group has also made significant strides in this area, developing 3D COFs with diverse networks using either 6- or 8-connected building units[36–40], such as the super-large porous JUC-564 with **stp** net[36], the non-interpenetrated hea net exhibited by JUC-596[37], and the mesoporous JUC-589 with a **bcu** net[38]. Despite these achievements, it is important to note that the structural diversity of existing 3D COFs pales in comparison to the vast repertoire of nodes available in the Reticular Chemistry Structure Resource (RCSR) database, which encompasses over 2000 networks with up to 24 extension points[41]. Therefore, expanding the structural variety of 3D COFs represents an imminent yet formidable challenge that warrants further investigation and innovative approaches.

Here, we present the discovery of two −3D COFs, named JUC-641 and JUC-642 (JUC = Jilin University China), which possess a **nia** topology. What sets these materials apart is the introduction of planar hexagonal and triangular prism nodes, and thus these COFs exhibit exceptional properties, including high crystallinity, a through-hole framework, good thermal and chemical stability, and a high specific surface area. Remarkably, our adsorption studies and breakthrough experiments showcase the high separation capabilities of both COFs. They outperform previously reported 3D COFs and most of porous organic polymers (POPs), with an impressive separation factor of up to 2.02 for benzene (Bz) and cyclohexane (Cy). Furthermore, our analysis utilizing dispersion-corrected density functional theory (DFT-D) lends insight into the performance of these 3D COFs. It suggests that the incorporation of highly aromatic building blocks and the presence of extensive pore structures contribute significantly to their separation properties.

## Results

### Structural design

By adhering to the guiding principles of reticular chemistry, it becomes possible to achieve rational regulation of topology through the careful selection of building units featuring different geometric shapes[3]. An example of this principle in action is the creation of a 2D COF with **hxl** topology, accomplished by combining planar hexagonal nodes (Fig. 1a)[42]. Similarly, the utilization of two triangular prism nodes has led to the successful synthesis of a 3D COF with **acs** topology (Fig. 1b)[39]. Through these studies, a clearer understanding of COF topology and its construction has been gained. In theory, assembling planar hexagons and triangular prism building blocks is expected to create 3D COFs with novel network structures such as **nia** (Fig. 1c), thereby exploring the structural diversity of COFs. However, thus far, no framework combining these two nodes has been explored. Building upon this concept, we have designed two 6-linked building blocks: 2,3,6,7,10,11-hexa (4′-aminophenyl)trimethylene (HAPTM, Fig. 1d) as a planar hexagonal node, and either 2,3,6,7,14,15-hexa(4′-formylphenyl)triptycene (HFPTP-H, Fig. 1e) or 2,3,6,7,14,15-hexa(3′-fluorine-4′-formylphenyl)triptycene (HFPTP-F) as trigonal prism nodes. The structure of HAPTM deviates from the typical graphitic geometries because of steric crowding between the hydrogens on the peripheral phenyl ring, which forces the aniline residues on the outer rim triphenylene planes to be alternately arranged up and down. The condensation of HAPTM with rigid prism HFPTP-H or HFPTP-F yields two expanded [6 + 6] junctional frameworks, designated as JUC-641 and JUC-642, respectively (Fig. 1f). Although different 3D topologies, such as **nia** and **htp** (Fig. 1g and Supplementary Fig. 4), are anticipated based on this combination, the resulting architecture primarily adopts a **nia** network because the periodic DFT calculation results show that the energy of COF with **nia** net is 71.7 kJ/unit lower than that of COF with **htp** net, indicating the **nia** net is energy preferred structure (see Supplementary Table 1).

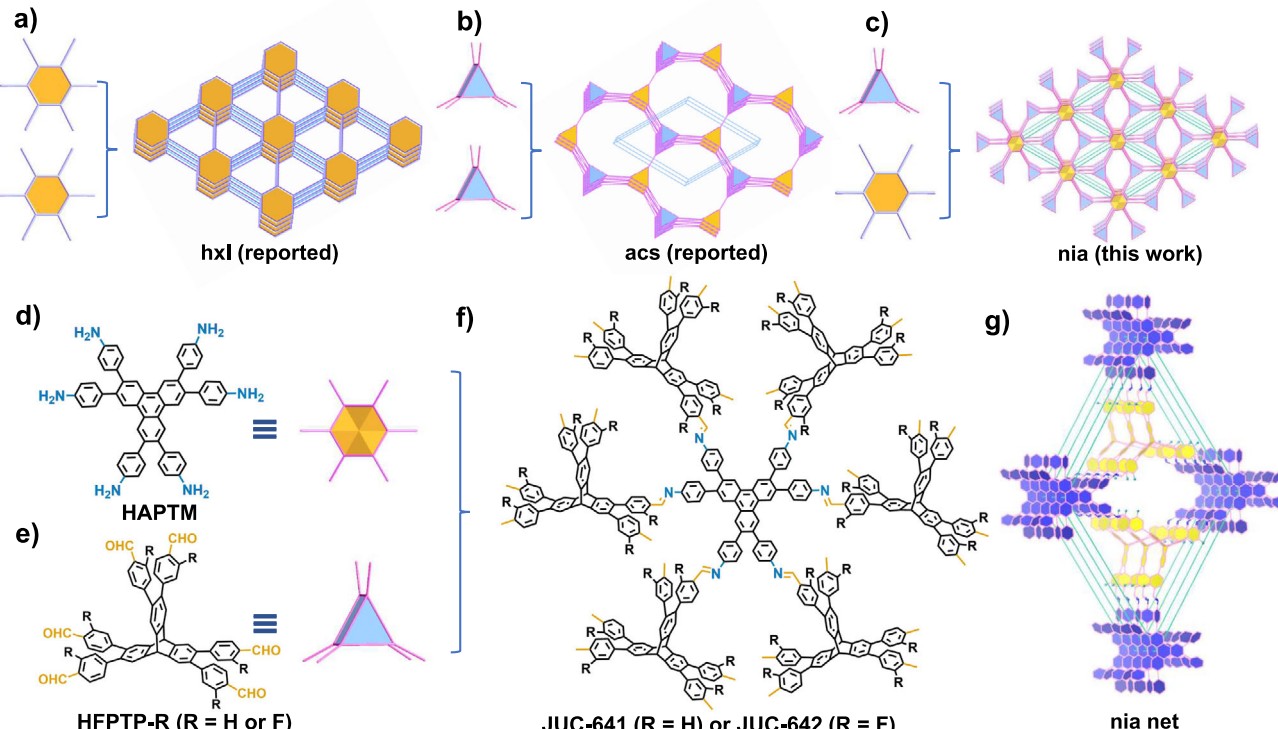

**Fig. 1 | A strategy for constructing JUC-641 and JUC-642 with a nia net.** The construction strategy of **hxl** a, **acs** b and **nia** c topology. Molecular structures of HFPTP-R (R = H or F, **d**) and HAPTM **e**. Two 3D COFs, JUC-641 and JUC-642, constructed from the condensation reaction of HFPTP-R and HAPTM **f**. An extended **nia** net **g** for both COFs.

## Synthesis and characterization

The syntheses of both COFs, JUC-641, and JUC-642, were conducted via solvothermal reaction of HFPTP-R (R = H or F) and TFAPM in a 10:1 (v/v) mixture of 1,4-dioxane/acetic acid at 120 °C for a duration of 7 days (Supplementary Section 1 in Supplementary information). The presence of new C = N stretching bands in the Fourier transform infrared (FT-IR) spectra (1620 cm$^{-1}$ for JUC-641 and 1623 cm$^{-1}$ for JUC-642) confirmed the successful polymerization of aldehyde and amine components (Supplementary Figs. 5 and 6). Additionally, solid-state $^{13}$C cross-polarization/magic-angle-spinning (CP/MAS) NMR spectra provided evidence for the formation of carbon atoms resulting from imine groups (161 ppm for JUC-641 and JUC-642, Supplementary Figs. 7 and 8). Notably, thermal-gravimetric analysis (TGA) revealed negligible weight loss for both COFs when subjected to heating up to 450 °C under a nitrogen atmosphere (Supplementary Figs. 9 and 10). Exceptional chemical stability was further confirmed by the preserved powder X-ray diffraction (PXRD) patterns observed after immersing the COFs in various organic solvents, boiling water, strong acid (3 M HCl), and strong base (3 M NaOH) for a period of 24 hours (Supplementary Figs. 11–14). Moreover, scanning electron microscopy (SEM) images indicated a uniform rod-like morphology for both COFs (Fig. 2c and d), while transmission electron microscopy (TEM) images exhibited their highly crystalline nature (Supplementary Figs. 15 and 16).

## Structural determination

The crystalline structures of JUC-641 and JUC-642 were confirmed using powder X-ray diffraction (PXRD) in combination with structural simulations (Fig. 2a, b). The simulated structures were generated using the Materials Studio software package (Supplementary Tables 10–13)[43]. Based on the geometry of the monomers, we constructed the frameworks of both COFs using the non-interpenetrated **nia** net from the RCSR with the *P-62c* space group (No. 190). Furthermore, Pawley refinement was employed for full profile pattern matching. Figure 1a displays the PXRD pattern of JUC-641, with prominent peaks observed at 3.42°, 5.23°, 6.05°, 6.78°, 8.01°, and 10.93°, corresponding to the (100), (110), (200), (101), (210), and (220) facets, respectively. A similar PXRD pattern (Fig. 2b) was obtained for JUC-642, with characteristic peaks at 3.42° (100), 5.26° (110), 6.07° (200), 6.81° (101), 8.02° (210), and 10.98° (220). The refinement results yielded comparable unit cell parameters for both COFs ($a = b = 30.3349$ Å,

$c = 22.8400$ Å, $\alpha = \beta = 90°$, and $\gamma = 120°$ for JUC-641; $a = b = 30.0413$ Å, $c = 23.4000$ Å, $\alpha = \beta = 90°$, and $\gamma = 120°$ for JUC-642) and good agreement factors ($Rp = 2.18\%$, $Rwp = 2.92\%$ for JUC-641; $Rp = 3.96\%$, $Rwp = 5.53\%$ for JUC-642). We also explored alternative structures with htp topology but observed significant discrepancies between the simulated and experimental PXRDs (Supplementary Figs. 17 and 18).

To further investigate the structures of both 3D COFs, we conducted high-resolution transmission electron microscopy (HRTEM) tests. Figure 2e–h reveals clear lattice fringes in the HRTEM images, with the spacing of 27.58, 18.47, 15.13, and 13.82 Å, observed for JUC-642, corresponding to the (100), (101), (110), and (200) Bragg peaks, respectively, which agree well with the simulated structure (Supplementary Figs. 19 and 20). These HRTEM images provide further support for the validity of the simulated structure for both COFs. Based on the aforementioned findings, we posit that both COFs conform to the anticipated 3D **nia** network, characterized by the presence of two distinct cavities. These cavities boast optimal dimensions of 1.0 nm along the *a* or *b* axis, as well as 1.5 nm along the *c* axis, as depicted in Supplementary Fig. 23.

## Adsorption test

The permanent porosity and specific surface areas of both COFs were assessed through N$_2$ adsorption measurements at 77 K. Figure 3 illustrates the isotherms of both COFs, which display a sharp uptake at a low pressure of P/P$_0$ < 0.05, followed by a gentle incline in the 0.05–0.2 P/P$_0$ range, indicating their microporous nature. Calculation of the Brunauer–Emmett–Teller (BET) specific surface areas yielded values of 1138 m$^2$ g$^{-1}$ for JUC-641 and 1610 m$^2$ g$^{-1}$ for JUC-642, as depicted in Supplementary Figs. 24–27. The lower BET surface area of JUC-641 can be attributed to its decreased crystallinity compared to JUC-642. Furthermore, analysis of the pore size distributions using non-local density functional theory (NLDFT) revealed two predominant pores with sizes of 1.11 and 1.47 nm for JUC-641 and 1.09 and 1.53 nm for JUC-642, aligning with the predicted pore sizes based on their crystal structures (Fig. 3).

## Breakthrough experiment

Moreover, considering the highly interconnected pore structure, optimal pore size dimensions, and abundant isolated π-systems present in JUC-641 and JUC-642, our research entailed the study of Bz and

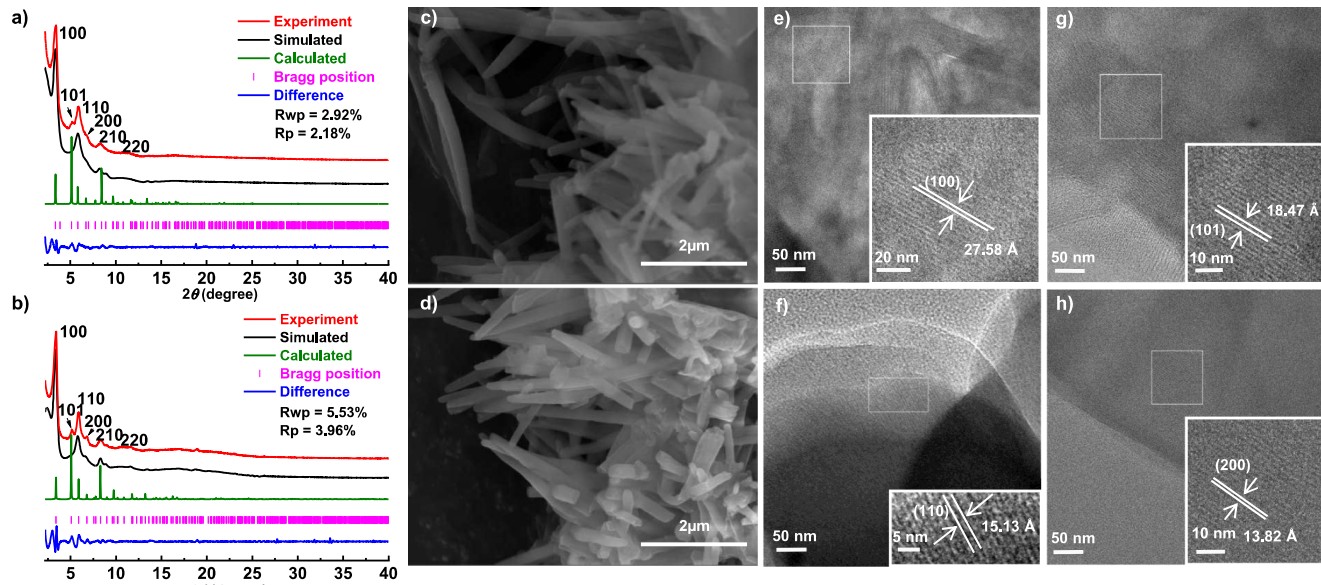

**Fig. 2 | Structural determination and characterization for both COFs.** PXRD patterns of JUC-641 **a** and JUC-642 **b**. SEM images of JUC-641 **c** and JUC-642 **d**. HRTEM images for JUC-642 (crystal face 100 **e**; 110 **f**; 101 **g**; 200 **h**).

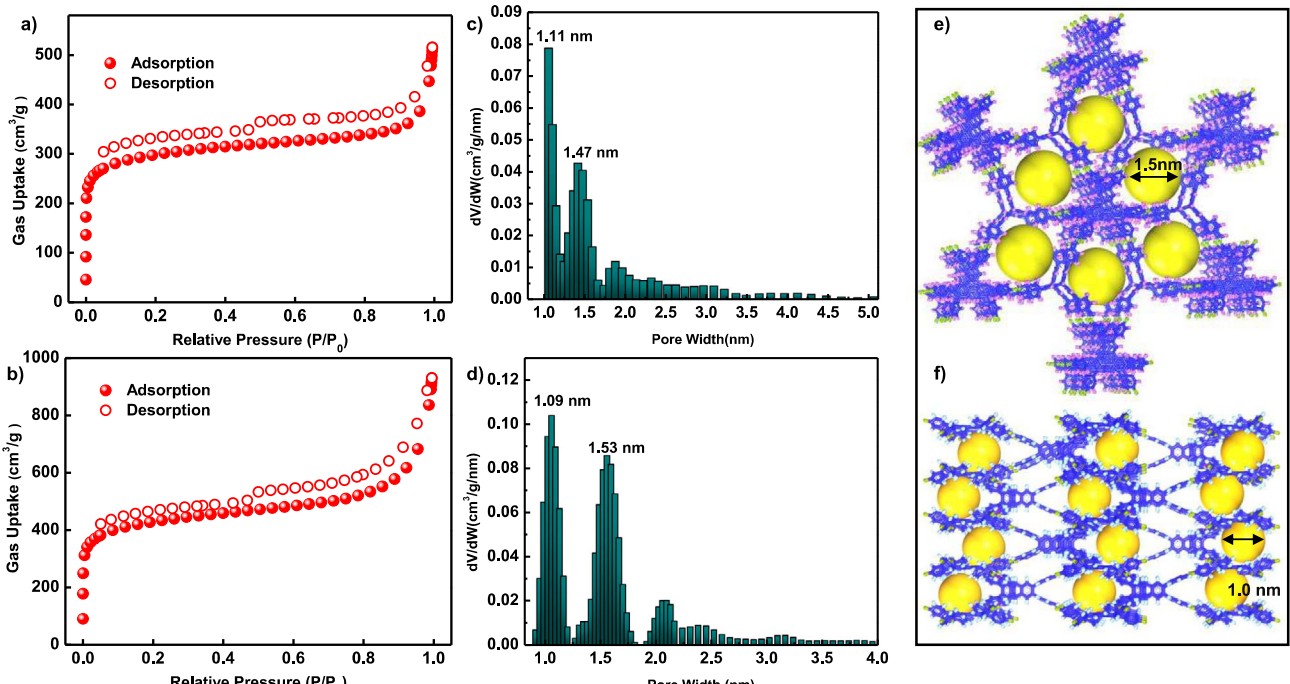

**Fig. 3 | Adsorption test and pore structure.** N₂ adsorption–desorption isotherms and their pore-size distributions of JUC-641 **a**, **c** and JUC-642 **b**, **d**. Expanded structures of JUC-641 observed along the *c* axis **e** and *a* or *b* axis **f**.

Cy vapor adsorption at atmospheric pressure and 298 K and 323 K, as well as breakthrough experiments for their subsequent separation (Fig. 4 and Supplementary Figs. 28 and 29). The determination of the isosteric heat of interaction ($Q_{st}$) is crucial for understanding the adsorption process of Bz and Cy in these frameworks. Results from the virial equation[44] reveal at zero coverage higher $Q_{st}$ values for Bz (36.6 kJ/mol for JUC-641 and 39.7 kJ/mol for JUC-642, Supplementary Fig. 30) compared to Cy (20.4 kJ/mol for JUC-641 and 23.3 kJ/mol for JUC-642, Supplementary Fig. 31), indicating a stronger interaction and higher adsorption affinity of Bz for both materials. This selectivity in favor of Bz suggests the potential utility of both materials in separating aromatic compounds in gas-phase applications. The separation of Bz and Cy holds significant importance due to Bz's role as a vital petrochemical product and Cy's crucial usage in resin and nylon fiber production[45,46]. Presently, commercially prevalent methods for separating Bz/Cy mixtures involve extractive distillation and azeotropic distillation. Nonetheless, these conventional approaches are both costly and energy-intensive. In order to address this issue, adsorptive separation methods employing porous adsorbents have garnered increased attention, owing to their superior separation efficiency and lower energy consumption compared to conventional approaches[47,48]. Therefore, our investigations seek to broaden the potential of 3D COFs as cutting-edge porous adsorbents in the field of petrochemical feedstock separation.

The Bz and Cy vapor adsorption/desorption isotherms of JUC-641 and JUC-642, depicted in Fig. 4a, b, reveal a notable disparity in the uptake between Bz and Cy vapors. This behavior can be attributed to two key factors. Firstly, the presence of abundant aromatic phenyl groups in the COF structure provides a greater number of sites with a strong affinity for the aromatic Bz molecules due to the robust π-π interaction. Consequently, the nonaromatic Cy molecules are more prone to being less retained. Secondly, the vapor adsorption capacity of the COF is significantly influenced by the critical temperature (Tc) of the molecules[49]. In this case, Bz possesses a higher critical temperature (Tc = 562.05 K) compared to Cy (Tc = 553.50 K). At 298 K and 323 K, Bz exhibits a higher propensity for adsorption and condensation within

the pores. Hence, the Tc of molecules is a key factor in determining the vapor adsorption capacity of COFs.

At a relative pressure of 0.95 (P/P₀), the Bz vapor uptake observed for JUC-641 and JUC-642 reached values of 566 mg g⁻¹ and 582 mg g⁻¹, respectively. These values can be compared to those of most POPs, such as PAF-2 (138 mg g⁻¹)[50], PCN-TPC (176 mg g⁻¹)[51], and MALP-1 (585 mg g⁻¹)[52]. Similarly, under equivalent temperature and relative pressure conditions, the Cy vapor uptake of JUC-641 and JUC-642 was 293 mg g⁻¹ and 287 mg g⁻¹, respectively. Importantly, the Bz adsorption capacity of JUC-641 and JUC-642 is 1.93 and 2.02 times greater than that of Cy, respectively. These values exceed the ideal Bz/Cy selectivity of previously reported 3D COFs, such as LZU-111 (1.21), COF-300-st (1.54), and COF-300-rt (1.33, Supplementary Table 2)[53], making them amongst the highest-selectivity materials in the field of POPs (Supplementary Table 3). Furthermore, breakthrough experiments were conducted with equimolar Bz and Cy mixtures for JUC-641 and JUC-642 at 298 K (Supplementary Fig. 32). As depicted in Fig. 4c, d, it demonstrates the sequential traversal of Cy vapor through the chromatographic column, displaying breakthrough times of 631 s g⁻¹ for JUC-641 and 531 s g⁻¹ for JUC-642. Conversely, for Bz, the breakthrough times are 1058 s g⁻¹ for JUC-641 and 923 s g⁻¹ for JUC-642. This disparity in breakthrough times can be attributed to the lower adsorption affinity between the cycloaliphatic Cy vapor and the COF skeleton. Based on these findings, the adsorption and diffusion coefficients of Bz and Cy, as well as the actual Bz/Cy selectivity, are presented in Supplementary Tables 4–9. Notably, these two COFs exhibit a stronger affinity for Bz compared to Cy, evident from their significantly higher Bz capacities and the much smaller Bz diffusion coefficient observed in the breakthrough experiment when compared to that of Cy. Importantly, the real Bz/Cy selectivity of the two COFs, 1.80 for JUC-641 and 1.91 for JUC-642, aligns with the ideal selectivity depicted in Fig. 4e, f, marking the first instance of Bz/Cy separation through a breakthrough experiment in COF research. Subsequently, a desorption experiment was conducted using argon gas (50 ml/min) blowing at 373 K after the packed bed reached saturation. As expected, due to the low binding affinity of JUC-642 to Cy, the adsorbed Cy is desorbed faster than Bz

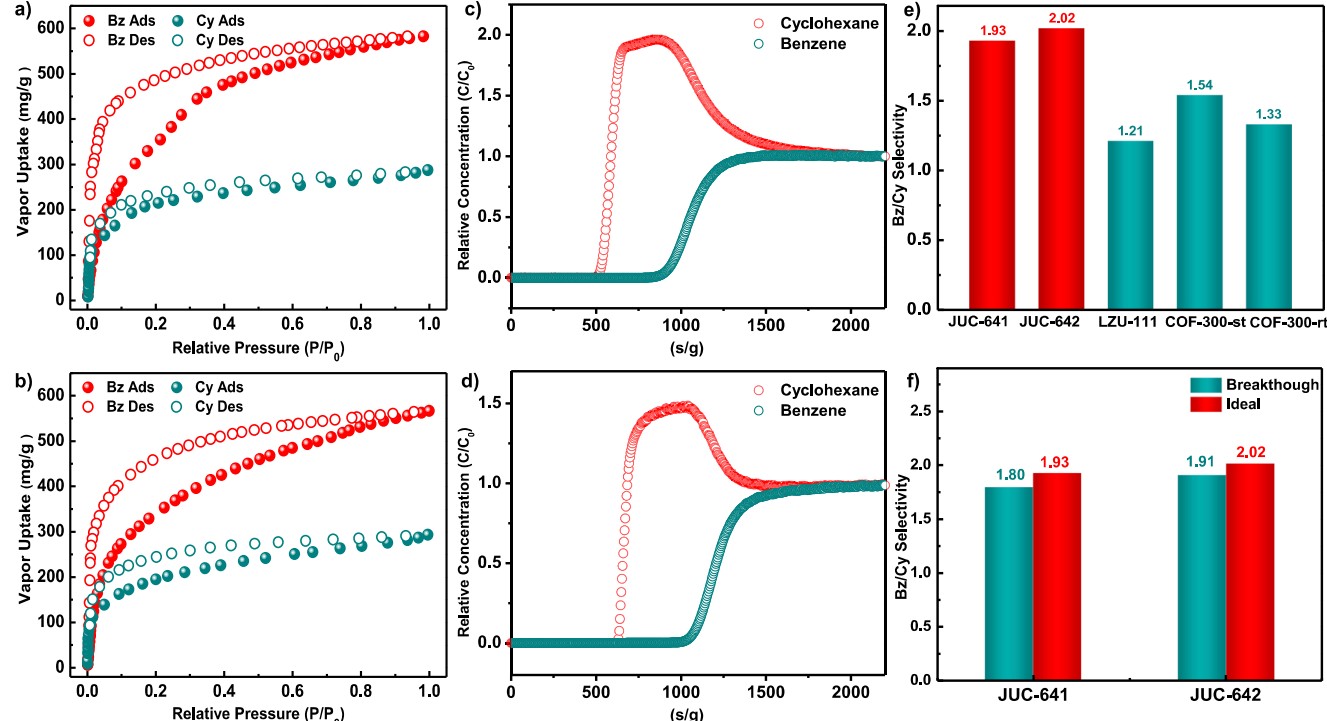

**Fig. 4 | Vapor adsorption and breakthrough experiment for both COFs.** Vapor adsorption (filled)–desorption (empty) isotherms of Bz (benzene) and Cy (cyclohexane) for JUC-641 **a** and JUC-642 **b** at 298 K. Breakthrough experiments of JUC-641 **c** and JUC-642 **d** with an equimolar mixture of Bz and Cy at 298 K. A comparison diagram of the ideal Bz/Cy selectivity for various 3D COFs **e**. A comprehensive analysis of the ideal and experimental selectivity of JUC-641 and JUC-642 in separating Bz/Cy at 298 K **f**.

(Supplementary Fig. 33). In our exploration of Bz and Cy interaction with JUC-642, a Chemisorption-Temperature Programmed Desorption (TPD) test at a gradual heating rate of 5 K/min was conducted. The desorption characteristics, revealed in Supplementary Figs. 34 and 35, show a significant temperature difference between Bz (110.6 °C) and Cy (86.0 °C), despite their similar boiling points (80.1 °C for Bz and 80.7 °C for Cy). This observed temperature gap provides empirical evidence supporting the stronger adsorption affinity of Bz to JUC-642.

### DFT-D calculations
To gain deeper insights into the factors underlying the disparities in adsorption and separation between JUC-641 and JUC-642 with Bz and Cy, we employed DFT-D to computationally determine the interaction energies between COF fragments and the adsorbates at three distinct locations. As depicted in Fig. 5, the interaction energy between the triphenylene unit within the COF and Cy (−22.48 kJ/mol) is notably lower compared to that with Bz (−28.02 kJ/mol). In the case of JUC-641, the interaction energies between the adjacent phenyl groups of the triptycene moiety and Bz (−51.24 kJ/mol) and Cy (−50.12 kJ/mol) are relatively similar. Conversely, the interaction energy between the dipyramidal region of the triptycene and Bz (−38.73 kJ/mol) is lower than that with Cy (−45.21 kJ/mol). On the other hand, for JUC-642, the interaction energies between Bz and the adjacent phenyl groups and dipyramidal region of the triptycene moiety (−40.17 kJ/mol and −47.00 kJ/mol, respectively) are stronger compared to those with Cy (−33.41 kJ/mol and −44.86 kJ/mol, respectively). Additionally, it is worth noting that aromatic Bz exhibits a stronger tendency to form multimers in nanoporous materials due to the π-π accumulation and confinement effect of pore structures. Consequently, Bz becomes the dominant molecule in the competitive adsorption process. By optimizing the geometry, we further confirmed that the distance between the hydrogen of C-H in the adsorbate and the nearest aromatic ring center of the skeleton is in the range of 2.72-4.93 Å, indicating that these geometries favor the formation of C-H...π interactions between the guest and the electron-rich backbone (Supplementary Fig. 36). The computational analysis reveals that Bz demonstrates a stronger affinity for the adsorption sites compared to Cy, providing an explanation for the adsorption and separation tendencies exhibited by JUC-641 and JUC-642 towards these two compounds. Moreover, the presence of fluorine atoms on the COF fragments notably enhances the interaction between the framework and the guest molecules, establishing JUC-642 as the more favorable choice for separation applications. This computational analysis provides theoretical evidence supporting the stronger adsorption affinity of Bz to these COF materials, complementing the observed temperature gap and $Q_{st}$ values. The multidimensional approach, combining theory and experimentation, enhances our understanding of intricate adsorption behaviors in porous materials.

### Discussion
In summary, we have successfully synthesized two distinct 3D COFs, JUC-641 and JUC-642, employing the integration of planar hexagonal and triangular prism building units. These COFs exhibit characteristics such as high crystallinity, exceptional thermal and chemical stability, and a well-defined porous structure. Notably, both COFs demonstrate remarkable performance in the separation of Bz and Cy, with JUC-641 reaching an optimal Bz/Cy selectivity of 1.93 and JUC-642 achieving a selectivity of 2.02. Encouragingly, breakthrough experiments reveal that the actual Bz/Cy selectivity of these COFs aligns quite closely with the ideal selectivity. Theoretical investigations conducted using DFT-D further support our findings by providing additional evidence of enhanced interactions between the Bz molecules and the COF framework. Thus, this study not only expands the repertoire of 3D COF architectures but also establishes functional COF materials as promising candidates for environmentally-friendly separation applications in the petrochemical industry.

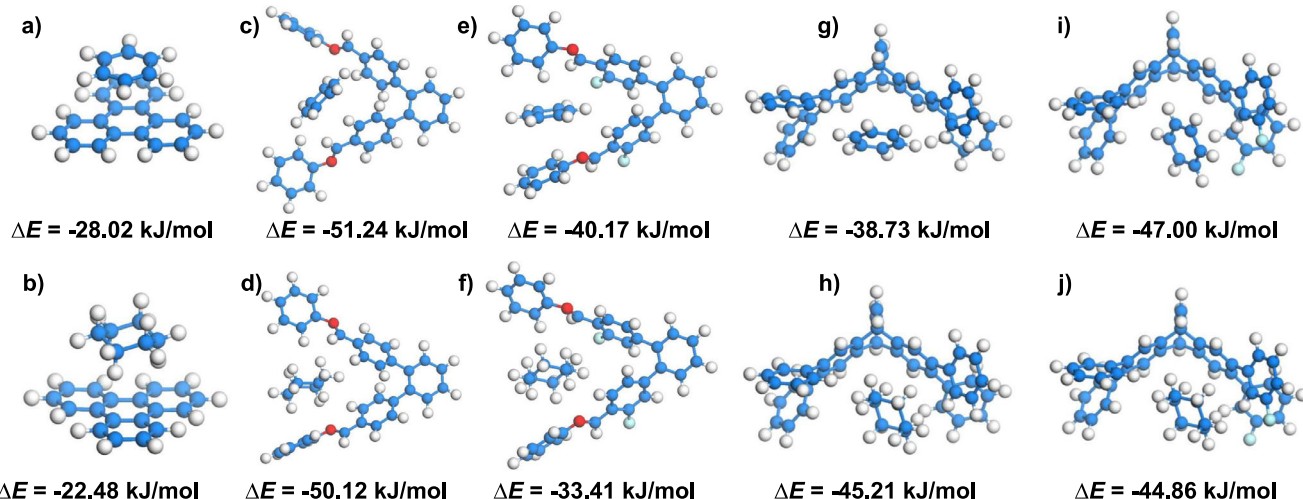

$\Delta E = -28.02$ kJ/mol   $\Delta E = -51.24$ kJ/mol   $\Delta E = -40.17$ kJ/mol   $\Delta E = -38.73$ kJ/mol   $\Delta E = -47.00$ kJ/mol

$\Delta E = -22.48$ kJ/mol   $\Delta E = -50.12$ kJ/mol   $\Delta E = -33.41$ kJ/mol   $\Delta E = -45.21$ kJ/mol   $\Delta E = -44.86$ kJ/mol

**Fig. 5 | Theoretical calculation.** Interaction energy ($\Delta E$) between the triphenylene unit with Bz **a** and Cy **b**. The adjacent phenyl groups of the triptycene moiety in JUC-641 with Bz **c** and Cy **d**. The adjacent phenyl groups of the triptycene moiety in JUC-642 with Bz **e** and Cy **f**. The dipyramidal region of the triptycene in JUC-641 with Bz **g** and Cy **h**. The dipyramidal region of the triptycene in JUC-642 with Bz **i** and Cy **j**.

## Methods

All available starting compounds and solvents, unless otherwise noted, were obtained from J&K Scientific LTD and used without further purification. All products were isolated and performed under nitrogen using either glovebox or Schlenk line techniques.

### Synthesis of JUC-641

HFPTP-H (0.016 mmol, 14.0 mg) and HBATP (0.016 mmol, 12.4 mg) were loaded into a Pyrex tube, 2.0 mL 1,4-dioxane and 0.2 mL of acetic acid (6 M) was added. The liquid nitrogen bath was used to freeze the Pyrex tube, which was evacuated to an interior pressure of ca. 19.0 mbar and flame-sealed, decreasing the whole length by ca. 10.0 cm. The tube was placed in an oven at 120 °C for 7 d after the temperature increasing to room temperature. The resulting precipitate was filtered, exhaustively washed with acetone for 5 times. The obtained powder was immersed in anhydrous n-hexane, and the solvent was exchanged with fresh n-hexane for several times. The sample was then transferred to vacuum chamber and evacuated to 20 mTorr under 65 °C, yielding white powder for $N_2$ adsorption measurements.

### Synthesis of JUC-642

HFPTP-F (0.015 mmol, 15.0 mg) and HBATP (0.015 mmol, 11.6 mg) were loaded into a Pyrex tube, 1.0 mL 1,4-dioxane and 0.1 mL of acetic acid (6 M) was added. The liquid nitrogen bath was used to freeze the Pyrex tube, which was evacuated to an interior pressure of ca. 19.0 mbar and flame-sealed, decreasing the whole length by ca. 10.0 cm. The tube was placed in an oven at 120 °C for 7 d after the temperature increasing to room temperature. The resulting precipitate was filtered, exhaustively washed with acetone for 5 times. The obtained powder was immersed in anhydrous n-hexane, and the solvent was exchanged with fresh n-hexane for several times. The sample was then transferred to vacuum chamber and evacuated to 20 mTorr under 65 °C, yielding white powder for $N_2$ adsorption measurements.

### Characterization

A Bruker AV-400 NMR spectrometer was applied to record the liquid 1H NMR spectra. Solid-state 13C NMR spectra were recorded on an AVIII 500 MHz solid-state NMR spectrometer. The FTIR spectra (KBr) were obtained using a SHIMADZU IRAffinity-1 Fourier transform infrared spectrophotometer. Thermogravimetric analysis (TGA) was recorded on a SHIMADZU DTG-60 thermal analyzer under $N_2$. PXRD data were collected on a PANalytical B.V. Empyrean powder diffractometer using a Cu Kα source ($\lambda = 1.5418$ Å). For scanning electron microscopy (SEM) images, JEOL JSM-6700 scanning electron microscope was applied. The transmission electron microscopy (TEM) images were obtained on JEM-2100 transmission electron microscopy.

### Nitrogen adsorption-desorption isotherms

The sorption isotherm for $N_2$ was measured by using a Quantachrome Autosorb-IQ analyzer with ultra-high-purity gas (99.999% purity). Before gas adsorption measurements, the as-synthesized COFs (~50.0 mg) were immersed in DMF for 12 h (3 × 5.0 mL) and then acetone for another 36 h (3 × 5.0 mL). The acetone was then extracted under vacuum at 85 °C to afford the samples for sorption analysis. To estimate pore size distributions for COFs, nonlocal density functional theory (NLDFT) was applied to analyze the $N_2$ isotherm on the basis of the model of $N_2$@77 K on carbon with slit pores and the method of non-negative regularization.

### Multi-constituent adsorption breakthrough curves

Multi-constituent Adsorption Breakthrough Curve were measured by BSD-MAB instrument. The breakthrough curves of 50:50 v/v mixture of Bz/Cy were collected at 298 K taking advantage of the oven of a gas chromatograph.

## Data availability

The authors declare that the data supporting the findings of this study are available within the Article and its Supplementary Information files, or from the corresponding author on request.

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

## Acknowledgements

This work was supported by National Key R&D Program of China (2022YFB3704900 and 2021YFF0500500), National Natural Science Foundation of China (22025504, 21621001, and 22105082), the SINOPEC Research Institute of Petroleum Processing, "111" project (BP0719036 and B17020), China Postdoctoral Science Foundation (2023M741326, 2023Q0128), and the program for JLU Science and Technology Innovative Research Team. V.V., Q.F., and S.Q. acknowledge the collaboration in the Sino-French International Research Network "Zeolites" framework.

## Author contributions

Q.F. and H.L. were responsible for the overall design, direction and supervision of the project. J.C. and F.C. performed the experimental work. Z.W. and L.Z. took TEM images. H.Z. and J.Z. helped with the TGA and PXRD tests. J.S. was in charge of other physical measurements. V.V. and S.Q. discussed the results and contributed to the writing of the manuscript.

## Competing interests

The authors declare no competing interests.
