## [Peer Review File · Nature Communications]

Three-Dimensional Covalent Organic Frameworks with nia
Nets for Efficient Separation of Benzene/Cyclohexane
MixturesReviewers' Comments:

Reviewer #1:

Remarks to the Author:

The paper presents the discovery, design, synthesis, characterization, and performance evaluation of two novel 3D Covalent Organic Frameworks (COFs), JUC-641 and JUC-642. The paper presents a novel approach to COF design by integrating planar hexagonal and triangular prism building units, resulting in the discovery of two unique 3D COFs with a nia topology. This innovative approach contributes to the expanding structural diversity of COFs. However, 3D COFs are well known, and the authors did not provide any justification for choosing these specific COFs. The triptycene building block is an excellent addition. However, 1d CONTs and POPs have been made from a triptycene unit [please see *Angew. Chem. Int. Ed.*, 2023, e202300652; *Adv. Mater.*, 2023, 2209919; *J. Am. Chem. Soc.*, 2022, 144, 16052; *Nat. Chem.*, 2022, 14, 507]. So, how exactly do these 3-D COFs bring in the novelty? This aspect could be more evident at the moment.

The authors extensively characterize the synthesized COFs using various techniques, including Fourier transform infrared (FT-IR) spectroscopy, solid-state ¹³C cross-polarization/magic-angle-spinning (CP/MAS) NMR, thermal gravimetric analysis (TGA), powder X-ray diffraction (PXRD), scanning electron microscopy (SEM), transmission electron microscopy (TEM), and high-resolution transmission electron microscopy (HRTEM). This thorough characterization provides robust evidence for the structural features and crystallinity of the COFs. However, assessing how authors could figure out the topology is complicated. The PXRD pattern is quite broad. How could they index the peaks?

The BET surface area is relatively modest. Any reason?

While the structural design of the COFs is innovative, some parts of the text related to the design, assembly of building blocks, and conversion of these building blocks into COFs are complex and may require more straightforward explanations. Breaking down the process and highlighting the key steps could enhance reader comprehension.

It would be beneficial to include a discussion of potential limitations or challenges associated with the proposed approach, such as scalability of synthesis, potential impurities, or stability under various conditions. This would provide a more balanced view of the presented work.

Comparative Analysis: Although the separation performance is compared against other reported 3D COFs and porous organic polymers (POPs), it would be valuable to provide a more comprehensive comparison with existing separation methods (e.g., extractive distillation, azeotropic distillation) to contextualize the performance of the COFs better.

Reviewer #2:

Remarks to the Author:

In this work, the authors reported two novel 3D COFs (namely JUC-641 and JUC-642) with nia topology. These 3D COFs exhibit excellent separation ability toward the mixture of benzene and cyclohexane. The material possesses high crystallinity, exceptional thermal and chemical stability, and a well-defined porous structure. These COFs are expected to be a promising candidate for environmentally-friendly separation applications. The authors also elucidated deeper insights into the factors underlying the disparities in adsorption and separation between JUC-641 and JUC-642 with benzene and cyclohexane using DFT-D calculation. These findings are of great significance for further designing novel material in the environmentally-friendly separation applications. Overall, the manuscript is well-organized and suitable for the wide readership of *Nature Communications*. I suggest the publication of this work after addressing the following issues.

(1) There are several grammatical errors and spelling errors that appear in the manuscript, please check the full text carefully and make modifications.

(2) In introduction part, some 3D COF materials with different topologies were listed, but the cited literature is not comprehensive. 3D COFs with novel topologies reported by other research groups should also be properly enumerated (e.g. *J. Am. Chem. Soc.* 2022, 144, 40, 18511–18517).

(3) Please provide ¹H NMR spectra of synthesized monomers in the Supplementary Information section.

(4) In Supplementary Figure 2 and Supplementary Section 1, the abbreviation "HBATP" should be corrected as "HAPTM" for 2,3,6,7,10,11-hexa(4'-aminophenyl)trimethylene, so as to be consistent with the main body of the manuscript.

Reviewer #3:

Remarks to the Author:

The authors report the use of prism and hexagonal building blocks to yield 3D nia structures and studied their suitability for benzene/cyclohexane separation. The work is interesting but some additional experiments need to be carried out before being suitable for publication.

1) The cell parameters have been obtained from XRPD. The authors have compared these values with HRTEM images. The XRPD and HRTEM values clearly disagree. The authors should explain.

2) The BET Surface area and pore volume need to be compared with the ones expected from the structural model.

3) The authors have calculated the Bz and Cy interaction energy from DFT studies. It will be good if they can compare these calculated data from experimental ones from registered the benzene and cyclohexane at two different temperatures.

4) The authors have calculated a Bz/Cy selectivity value from breakthrough curves. The shape of Cy breakthrough clearly shows, in the eluted flow, an increase above the initial concentration of Cy which is indicative of a displacement of adsorbed Cy by Bz. In this regard study of the adsorbed phase should be made (i.e. temperature programmed desorption or by adsorbate phase analysis by extraction and NMR analysis).

The term "absorption" should be replaced by adsorption.

Below is a point-by-point response to the comments of reviewers and the editor and the corresponding amendments in the revised manuscript and Supplemental Information.

Reviewer #1:

(1) The paper presents the discovery, design, synthesis, characterization, and performance evaluation of two novel 3D Covalent Organic Frameworks (COFs), JUC-641 and JUC-642. The paper presents a novel approach to COF design by integrating planar hexagonal and triangular prism building units, resulting in the discovery of two unique 3D COFs with a nia topology. This innovative approach contributes to the expanding structural diversity of COFs. However, 3D COFs are well known, and the authors did not provide any justification for choosing these specific COFs. The triptycene building block is an excellent addition. However, 1d CONTs and POPs have been made from a triptycene unit [please see *Angew. Chem. Int. Ed.*, 2023, e202300652; *Adv. Mater.*, 2023, 2209919; *J. Am. Chem. Soc.*, 2022, 144, 16052; *Nat. Chem.*, 2022, 14, 507]. So, how exactly do these 3-D COFs bring in the novelty? This aspect could be more evident at the moment.

*Response: We thank the reviewer for this insightful comment, and these important references about 1d CONTs and POPs made from triptycene units (*Angew. Chem. Int. Ed.*, 2023, e202300652; *Adv. Mater.*, 2023, 2209919; *J. Am. Chem. Soc.*, 2022, 144, 16052; *Nat. Chem.*, 2022, 14, 507) have been discussed and cited in the revised manuscript (please see pages 5,6 and Refs. 44-47). Triptycene has a unique rigid 3D scaffold structure, which can prevent π - π stacking, endow the materials with high void space, and is conducive to optimizing the pore channels and expanding the space of the materials. The development of POPs and 1D CONTs based on triptycene units has made significant progress in advanced materials with enhanced structural and functional properties, but they are usually amorphous and cannot accurately determine the pore structure. 3D COFs offer distinct advantages over amorphous porous materials due to their precisely tailored structures and high crystallinity, providing exceptional thermal stability and superior separation capabilities, making them promising candidates for various applications. Expanding triptycene into 3D COF materials is of great significance in clarifying the unique pore structure characteristics brought by triptycene and improving the various properties. Since we introduced the triptycene unit into 3D COFs, our research group has conducted numerous groundbreaking studies (e.g., *J. Am. Chem. Soc.* 142, 13334-13338 (2020); *J. Am. Chem. Soc.* 143, 2654-2659 (2021); *Angew. Chem. Int. Ed.* 61, e202117101 (2022)).*

In this manuscript, we introduce the discovery of two novel 3D COFs, named JUC-641 and JUC-642, which possess a unique nia topology for the first time. What sets these materials apart is the incorporation of planar hexagonal and triangular prism nodes, resulting in exceptional properties such as high crystallinity, remarkable thermal and chemical stability, and a high specific surface area. Notably, our adsorption studies and breakthrough experiments demonstrate the high separation

capabilities of both COFs, surpassing previously reported 3D COFs and the majority of POPs with an impressive separation factor of up to 2.02 for benzene (Bz) and cyclohexane (Cy). Furthermore, our analysis utilizing dispersion-corrected density functional theory (DFT-D) provides insights into the outstanding separation properties of these 3D COFs. It suggests that the inclusion of highly aromatic building blocks and the presence of extensive pore structures significantly contribute to their exceptional separation performance.

In summary, this work provides comprehensive insights into the novelty of 3D triptycene-based COFs by highlighting their unique structural features and exceptional separation performance. It also justifies the selection of these specific 3D COFs based on the innovative combination of building blocks and their contributions to COF diversity and separation capabilities.

(2) The authors extensively characterize the synthesized COFs using various techniques, including Fourier transform infrared (FT-IR) spectroscopy, solid-state ¹³C cross-polarization/magic-angle-spinning (CP/MAS) NMR, thermal gravimetric analysis (TGA), powder X-ray diffraction (PXRD), scanning electron microscopy (SEM), transmission electron microscopy (TEM), and high-resolution transmission electron microscopy (HRTEM). This thorough characterization provides robust evidence for the structural features and crystallinity of the COFs. However, assessing how authors could figure out the topology is complicated. The PXRD pattern is quite broad. How could they index the peaks?

Response: *We thank the reviewer for this important question. We used a combination of experimental powder X-ray diffraction (PXRD) and structural simulation to verify the crystal structures of JUC-641 and JUC-642, which is currently the most widely used structural analysis method in 3D COFs. The specific steps are as follows: Firstly, we determine the possible topological structure through the RCSR website based on the geometric features of the selected building units, and use the Materials Studio software package to generate the corresponding simulation structure. Then, by comparing the simulated XRD data with the experimental PXRD data, we found that the XRD values of JUC-641 and JUC-642 simulated by the non-interpenetrated *nia* network with P-62c space group (No. 190) matched well with the experimental values. Subsequently, we refined and fine-tuned the parameters of the simulated structure through Pawley refinement to make it more consistent with the actual structure. By carefully searching the PXRD spectra of JUC-641 and JUC-642, we found prominent Bragg peaks corresponding to different planes within the lattice at specific diffraction angles; For example, significant peaks were observed at 3.42 °, 5.23 °, 6.05 °, 6.78 °, 8.01 °, and 10.93 ° in JUC-641, corresponding to (100), (110), (200), (101), (210), and (220) crystal planes, respectively. JUC-642 obtained a similar PXRD pattern. To further verify the accuracy of the *nia* topology, we also explored alternative structures with *htp* topology. However, we observed significant differences between simulated and experimental PXRD. In addition, we also conducted high-resolution transmission electron microscopy (HRTEM) testing. The HRTEM image shows clear lattice stripes with specific spacing, corresponding to the Bragg peak observed in the PXRD pattern.*

These HRTEM results provide further validation for the simulated structure.

(3) The BET surface area is relatively modest. Any reason?

Response: We thank the reviewer for this careful observation. The relatively modest BET surface area could be attributed to the insufficient level of crystallinity observed in the COF materials.

(4) While the structural design of the COFs is innovative, some parts of the text related to the design, assembly of building blocks, and conversion of these building blocks into COFs are complex and may require more straightforward explanations. Breaking down the process and highlighting the key steps could enhance reader comprehension.

Response: We sincerely appreciate the reviewer for providing this valuable suggestion. Taking the reviewer's advice into consideration, we have diligently reorganized the content pertaining to the structural design within the revised manuscript (please see pages 5,6 and Scheme 1).

(5) It would be beneficial to include a discussion of potential limitations or challenges associated with the proposed approach, such as scalability of synthesis, potential impurities, or stability under various conditions. This would provide a more balanced view of the presented work.

Response: We thank the reviewer for this suggestion. Exceptional chemical stability was further confirmed by the preserved powder X-ray diffraction (PXRD) patterns observed after immersing the COFs in various organic solvents, boiling water, strong acid (3 M HCl), and strong base (3 M NaOH) for a period of 24 hours (Supplementary Figs. 8-11). It is noteworthy that the achievement of highly crystalline COF materials necessitates the utilization of high-purity reaction precursors (HFPTP-R and TFAPM). Furthermore, increasing the quantity of these reaction precursors allows for the production of highly crystalline COF materials, thereby enabling the potential for large-scale synthesis of these materials. This related discussion has been provided in the revised manuscript, and please see page 7.

Reviewer #2:

In this work, the authors reported two novel 3D COFs (namely JUC-641 and JUC-642) with nia topology. These 3D COFs exhibit excellent separation ability toward the mixture of benzene and cyclohexane. The material possesses high crystallinity, exceptional thermal and chemical stability, and a well-defined porous structure. These COFs are expected to be a promising candidate for environmentally-friendly separation applications. The authors also elucidated deeper insights into the factors underlying the disparities in adsorption and separation between JUC-641 and JUC-642 with benzene and cyclohexane using DFT-D calculation. These findings are of great significance for further designing novel material in the environmentally-friendly separation applications.

Overall, the manuscript is well-organized and suitable for the wide readership of Nature Communications. I suggest the publication of this work after addressing the following issues.

Response: We greatly appreciate the positive comments on our results. We have addressed all of the reviewer's suggestions one by one as follows.

(1) There are several grammatical errors and spelling errors that appear in the manuscript, please check the full text carefully and make modifications.

Response: We thank the reviewer for the careful observation. We have carefully checked the entire manuscript and made necessary grammatical and spelling corrections.

(2) In introduction part, some 3D COF materials with different topologies were listed, but the cited literature is not comprehensive. 3D COFs with novel topologies reported by other research groups should also be properly enumerated (e.g. J. Am. Chem. Soc. 2022, 144, 40, 18511–18517).

Response: We thank the reviewer for this suggestion. We have updated the introduction section of the revised manuscript to include the important reference (J. Am. Chem. Soc. 2022, 144, 40, 18511–18517). (Please see page 3 and ref. 34)

(3) Please provide ^1H NMR spectra of synthesized monomers in the Supplementary Information section.

Response: We thank the reviewer for this suggestion. We have included the ^1H NMR spectra of the synthesized monomers in the revised Supplementary Information section. (Please see Supplementary Section I)

Figure R1: The ^1H NMR spectra of HFPTP-H.

Figure R2: The ^1H NMR spectra of HFPTP-F.

Figure R3: The ^1H NMR spectra of HAPTM.

(4) In Supplementary Figure 2 and Supplementary Section 1, the abbreviation “HBATP” should be corrected as “HAPTM” for 2,3,6,7,10,11-hexa(4'-aminophenyl)trimethylene, so as to be consistent with the main body of the manuscript.

Response: We thank the reviewer for the careful observation. We have corrected all the abbreviations from "HBATP" to "HAPTM" in the revised manuscript and Supporting Information.

Reviewer #3:

The authors report the use of prism and hexagonal building blocks to yield 3D nia structures and studied their suitability for benzene/cyclohexane separation. The work is interesting but some additional experiments need to be carried out before being suitable for publication.

Response: We greatly appreciate the positive comments on our results. We have addressed all of the reviewer's suggestions one by one as follows.

(1) The cell parameters have been obtained from XRPD. The authors have compared these values with HRTEM images. The XRPD and HRTEM values clearly disagree. The authors should explain.

Response: We thank the reviewer for the careful observation. Based on the formula for calculating crystal plane spacing in the hexagonal crystal system and using the refined cell parameters obtained through simulation (see Supplementary Section 9 and Table 10 in the Supplementary Information), we calculated the crystal plane spacing for (100), (101), (110), and (200) to be 2.602 nm, 1.740 nm, 1.502 nm, and 1.301 nm, respectively. These values slightly differ from those observed in high-resolution transmission electron microscopy (HRTEM) images.

In the hexagonal crystal system, the formula for calculating the interplanar spacing (d) is:

$$d_{hkl} = \frac{1}{\sqrt{\frac{4}{3} \left(\frac{h^2 + k^2 + hk}{a^2} \right) + \left(\frac{l}{c} \right)^2}}$$

Here, d represents the interplanar spacing, a and c is the lattice constant, and h , k , and l are the Miller indices of the crystal planes.

We attribute these discrepancies to a combination of multiple factors: (1) Simulation accuracy: Theoretical calculations rely on precise simulation parameters, which may introduce minor changes due to inherent limitations of modeling techniques. (2) Experimental variability: HRTEM measurements are influenced by experimental conditions and may involve inherent uncertainty, resulting in slight differences in the measured spacing. (3) Sample variability: Variations in the synthesized samples, such as slight differences in crystal quality or morphology, can impact the observed HRTEM results.

Overall, these factors collectively contribute to the minor differences observed in crystal plane spacing between theoretical calculations and HRTEM observations.

(2) The BET Surface area and pore volume need to be compared with the ones expected from the structural model.

Response: We thank the reviewer for this suggestion. We used the Zeo++ software to calculate the BET surface area and pore volume of the JUC-641 and JUC-642 structural models. The results showed that the BET surface areas of JUC-641 and JUC-642 were 6353 and 6021 m² g⁻¹, respectively. Additionally, the pore volumes of

JUC-641 and JUC-642 were 3.05 and 2.86 cm³ g⁻¹, respectively, which had a significant difference compared to the experimental values (JUC-641: 0.73 cm³ g⁻¹ and JUC-642: 1.33 cm³ g⁻¹). This difference can be attributed to the crystallinity and defects in the crystal structure. In the current field of 3D COFs, this phenomenon is commonly observed, where the actual BET surface areas of most 3D COFs range from 500 to 2000 m² g⁻¹, which significantly deviates from the theoretical values (Angew. Chem. Int. Ed. 61, e202204899 (2022); J. Am. Chem. Soc. 144, 5728–5733 (2022); Chem 8, 3064–3080 (2022); Nat. Commun. 14, 2865 (2023)).

(3) The authors have calculated the Bz and Cy interaction energy from DFT studies. It will be good if they can compare these calculated data from experimental ones from registered the benzene and cyclohexane at two different temperatures.

Response: *We thank the reviewer for this suggestion. We have conducted supplementary tests on the vapor adsorption-desorption of benzene and cyclohexane by JUC-641 and JUC-642 under 313 K conditions (please see pages 9,10 and Supplementary Figures 25,26 in the Supplementary Information). The experimental data shows that the adsorption of benzene vapor by both materials is higher than that of cyclohexane. This indicates that benzene has a higher affinity for active sites.*

Figure R4: Vapor adsorption (filled)–desorption (empty) isotherms of Bz and Cy for JUC-641 at 313 K.

Figure R5: Vapor adsorption (filled)–desorption (empty) isotherms of Bz and Cy for JUC-642 at 313 K.

(4) The authors have calculated a Bz/Cy selectivity value from breakthrough curves. The shape of Cy breakthrough clearly shows, in the eluted flow, an increase above the initial concentration of Cy which is indicative of a displacement of adsorbed Cy by Bz. In this regard study of the adsorbed phase should be made (i.e. temperature programed desorption or by adsorbate phase analysis by extraction and NMR analysis)

Response: We thank the reviewer for this suggestion. We studied the adsorption phase of JUC-642 through temperature program desorption. We directly transferred the saturated JUC-642 from penetration adsorption into a heating furnace, raising the temperature from room temperature to 100 degrees at a rate of 5 degrees per minute. Subsequently, we purged and desorbed the sample using argon gas at 373 K and a flow rate of 50 mL/min, while collecting mass spectrometry signals.

Figure R6: Desorption curves for Bz/Cy (v/v, 2/2) mixtures (Ar flow with 50 mL·min⁻¹ at 373 K).

The analysis results showed the coexistence of benzene and cyclohexane in the adsorption phase. Due to the low binding affinity of JUC-642 to cyclohexane, the adsorbed cyclohexane desorbed faster than benzene (please see page 11 and Supplementary Figure 28 in the Supplementary Information).

(5) The term “absorption” should be replaced by adsorption.

Response: *We thank the reviewer for the careful observation. We have corrected all spelling errors in the revised manuscript and Supporting Information.*

Reviewers' Comments:

Reviewer #1:

Remarks to the Author:

The authors have addressed all comments. This manuscript can be accepted now. I enjoyed reading this manuscript.

Reviewer #2:

Remarks to the Author:

The authors have revised the manuscript according to the comments of reviewers. The current manuscript has cleared all the related issues and is suitable for the wide readership of Nature Communications.

Reviewer #3:

Remarks to the Author:

The authors have only carried out some of the additional experiments requested by the reviewers. On top of that the answers to the reviewers questions do not meet in general the scientific quality criteria of a high quality journal.

The values for $a=b= 30.03 \text{ \AA}$ (PXRD) and 27.6 \AA (Electron diffraction) differ considerably. Moreover, while the reflections for PXRD are given in 2θ degrees the electron diffraction data are given in nm. The same units need to be given either in angstroms or nanometer. The answer that the differences may arise from different reasons is not scientifically appealing. Does it mean that the nanocrystals are not homogeneous and not similar to the bulk material (PXRD)?

The authors give only a qualitative answer to the higher strength of the benzene vs cyclohexane interaction. While the authors have measured the isotherms at two different temperatures why they have not at least calculated the value of isosteric heats of adsorption. These values should be calculated and discussed comparing them with the calculated from computational data. The desorption experiment at 313 K only gives information about the composition of the adsorbate phase. A new experiment using a heating ramp i.e. 5 K/min might give information about the differences in interaction for the two different adsorbate molecules.

Additional remarks: Caption to Figure 3 need to be corrected.

Below is a point-by-point response to the comments of Reviewer #3 and the corresponding amendments in the revised manuscript and Supplemental Information.

Reviewer #3:

The authors have only carried out some of the additional experiments requested by the reviewers. On top of that the answers to the reviewers questions do not meet in general the scientific quality criteria of a high quality journal.

Response: We greatly appreciate the valuable comments from the reviewer. In response to the feedback, we have made several improvements in the revised manuscript. Firstly, we have provided a detailed explanation for the discrepancy between the values of $a=b=30.03$ Å (PXRD) and 27.6 Å (Electron diffraction). Additionally, we have calculated the isosteric heats of adsorption and conducted a new experiment utilizing a heating ramp of 5 K/min. These updates address the concerns raised and enhance the scientific rigor of our study. We would like to express our gratitude for highlighting these issues, which have allowed us to improve the quality of our work. Please find below a more detailed account of our new findings and the corresponding explanations:

(1) The values for $a=b= 30.03$ Å (PXRD) and 27.6 Å (Electron diffraction) differ considerably. Moreover, while the reflections for PXRD are given in 2theta degrees the electron diffraction data are given in nm. The same units need to be given either amstrongs or nanometer. The answer that the differences may arise from different reasons is not scientifically appealing. Does it mean that the nanocrystals are not homogeneous and not similar to the bulk material (PXRD)?

Response: We thank the reviewer for the careful observation. The distinction in crystal systems plays a pivotal role in understanding the spatial arrangement of crystal planes and their respective interplanar distances. For example, in the tetragonal crystal system, a notable characteristic is observed: the separation between (100) crystal planes precisely aligns with the crystal cell parameter 'a'. This intrinsic relationship provides a straightforward correlation between the lattice parameters and the interplanar spacings in tetragonal structures (See Figure R1a).

Figure R1: Schematic diagram of interplanar spacing for the (100) plane in a) the tetragonal crystal system and b) the hexagonal crystal system, respectively.

However, when considering the hexagonal crystal system, unlike the straightforward correspondence seen in tetragonal systems, the interplanar distance between (100) crystal planes in hexagonal structures does not necessarily equate to the crystal cell parameter 'a' (See Figure R1b), as exemplified by the COF HFPTP-TAE with *stp* topology (J. Am. Chem. Soc. 2023, 145, 23227-23237), which has a distance of 41 Å between (100) crystal planes, while the crystal cell parameters are $a = b = 47.44$ Å.

$$d_{hkl} = \frac{1}{\sqrt{\frac{4}{3} \left(\frac{h^2 + k^2 + hk}{a^2} \right) + \left(\frac{l}{c} \right)^2}}$$

Figure R2. The formula for calculating the interplanar spacing (d_{hkl}) in the hexagonal crystal system.

Actually, both COFs in our example belong to the hexagonal crystal system. Therefore, we have utilized the formula, as illustrated in Figure R2, to compute the interplanar distances (d_{hkl}) of JUC-642. Leveraging the refined crystal cell parameters derived from our simulation efforts (see Supplementary Section 9 and Table 10), we calculated the interplanar distances for (100), (101), (110), and (200) to be 26.02, 17.40, 15.02, and 13.01 Å, respectively. These values are in good agreement with the values observed in high-resolution transmission electron microscopy (HRTEM) images, which are 27.58 for (100), 18.47 for (101), 15.13 for (110), and 13.82 Å for (200). Importantly, the minor disparities between the computed and observed values fall within a reasonable error range. These differences may be attributed to the inherent limitations of the modeling technique in simulating accuracy, the inherent uncertainty in HRTEM measurements, and the minor changes in crystal morphology during the measurement process. The alignment between calculated and observed interplanar distances, despite minor differences, underscores the robustness of our simulation approach. These findings contribute to the validation of our theoretical framework and provide valuable insights into the reliability of crystallographic modeling in capturing the intricate details of hexagonal crystal systems.

We thank the reviewer for this suggestion. We have unified the units for electron diffraction and crystal cell values (Figure 1 and page 7 in the revised manuscript and Supplementary Section 9). In addition, in terms of morphology as reflected by scanning electron microscopy (SEM) and transmission electron microscopy (TEM), both JUC-641 and JUC-642 exhibit a relatively uniform rod-like morphology (Figure 1 in the revised manuscript and Supplementary Figures 12,13).

(2) The authors give only a qualitative answer to the higher strength of the benzene vs cyclohexane interaction. While the authors have measured the isotherms at two different temperatures why they have not at least calculated the value of isosteric heats of adsorption. These values should be calculated and discussed comparing them with the calculated from computational

data. The desorption experiment at 313 K only gives information about the composition of the adsorbate phase. A new experiment using a heating ramp i.e. 5 K/min might give information about the differences in interaction for the two different adsorbate molecules.

Response: We sincerely thank the reviewer for these very valuable suggestions. The virial equation is utilized to calculate the isosteric heat of adsorption (Q_{st}) values for benzene and cyclohexane at near-zero loading. For JUC-641, the Q_{st} value was determined to be 36.6 kJ/mol for benzene and 20.4 kJ/mol for cyclohexane, respectively. On the other hand, higher Q_{st} values were observed for JUC-642, with 39.7 kJ/mol for benzene and 23.3 kJ/mol for cyclohexane (Figures R3, R4, Supplementary Figures 27,28, and pages 8,9 in the revised manuscript). The elevated Q_{st} values for benzene underscore its stronger interaction and higher adsorption affinity for both JUC-641 and JUC-642 compared to cyclohexane. Furthermore, the distinct Q_{st} values for benzene and cyclohexane within each material highlight the selectivity nature of the adsorption process.

Figure R3: Q_{st} of benzene in JUC-641 and JUC-642.

Figure R4: Q_{st} of cyclohexane in JUC-641 and JUC-642.

In addition, we have also utilized dispersion-corrected density functional theory (DFT-D) calculations to assess the interaction energies between distinct fragments of the COF structure and benzene as well as cyclohexane. The employment of DFT-D calculations offers valuable insights into the interaction dynamics under ideal conditions, providing a theoretical foundation for understanding the affinities between the framework structures and the two molecules. The computational analysis reveals that benzene demonstrates a stronger affinity for the adsorption sites compared to cyclohexane, providing an explanation for the adsorption and separation tendencies exhibited by JUC-641 and JUC-642 towards these two compounds. Therefore, the overarching consensus derived from both the DFT-D calculations and the experimentally determined Q_{st} values is consistent: benzene displays a heightened affinity for both JUC-641 and JUC-642 in comparison to cyclohexane. The multidimensional approach, combining theory and experimentation, enhances our understanding of intricate adsorption behaviors in porous materials. For more detailed information, please refer to pages 11 and 12 in the revised manuscript.

To deepen our exploration of the interaction between benzene and cyclohexane with the JUC-642, we conducted a Chemisorption-Temperature Programmed Desorption (TPD) test employing a gradual heating rate of 5 K/min, as illustrated in Figures R5, R6 (Supplementary Figures 31,32). This experimental approach enables us to scrutinize the desorption characteristics of the adsorbed molecules under controlled temperature conditions. A well-established in assessing the interaction strength between volatile organic compounds (VOCs) and adsorbents involves analyzing the difference between the maximum desorption temperature and the boiling point of the compounds at atmospheric pressure. This differential temperature provides a reliable indicator of the adsorption affinity, with a greater temperature disparity implying a stronger interaction (Environ. Sci.: Nano. 2022, 9, 81).

Figure R5: TPD of benzene for JUC-642.

Figure R6: TPD of cyclohexane for JUC-642.

In the JUC-642, the highest desorption peak for cyclohexane manifested around 86.0 °C, while the highest desorption peak for benzene occurred notably later, at 110.6 °C. It is noteworthy that the boiling points of benzene and cyclohexane are remarkably similar (80.1 °C for benzene and 80.7 °C for cyclohexane). The conspicuous discrepancy in desorption temperatures, with benzene desorbing at a significantly higher temperature than its boiling point, stands out as a compelling observation. This substantial temperature difference for benzene, exceeding that of cyclohexane, serves as a clear and tangible indication of the stronger adsorption affinity between benzene and the COF material (See page 11 in the revised manuscript). The experimental findings not only align with the outcomes from Q_{st} values but also provide an additional layer of empirical support for the higher affinity of benzene towards the COFs.

(3) Additional remarks: Caption to Figure 3 need to be corrected.

***Response:** We thank the reviewer for the careful observation. We have corrected error in the revised manuscript.*

Reviewers' Comments:

Reviewer #3:

Remarks to the Author:

The authors have addressed most of the concerns of reviewer 3 comments.

However, there is still one point that needs revision:

The values shown in Figure R3 (Qst of benzene in JUC-641 and JUC-642) at higher loadings do not seem reasonable. I believe that this issue is related to the non suitability of the virial method at high loadings.

Below is a point-by-point response to the comments of reviewers and the corresponding amendments in the revised manuscript and Supplemental Information.

Reviewer # 3:

The authors have addressed most of the concerns of reviewer 3 comments. However, there is still one point that needs revision: The values shown in Figure R3 (Q_{st} of benzene in JUC-641 and JUC-642) at higher loadings do not seem reasonable. I believe that this issue is related to the non-suitability of the virial method at high loadings.

Response: We appreciate the reviewer's comments and suggestions that helped us to significantly improve the manuscript. We agree with the reviewer's comments, and we have removed the data at high loadings in the revised Supplementary Figure 30.